# Spine Metastasis Is Associated with the Development of Brain Metastasis in Non-Small-Cell Lung Cancer Patients

**DOI:** 10.3390/medicina60010152

**Published:** 2024-01-14

**Authors:** Hyung-Keun Cha, Woo-Kyung Ryu, Ha-Young Lee, Hyun-Jung Kim, Jeong-Seon Ryu, Jun-Hyeok Lim

**Affiliations:** 1Department of Pulmonology, Internal Medicine, Inha University Hospital, Inha University College of Medicine, Incheon 22332, Republic of Korea; silmiboy@naver.com (H.-K.C.); ryuwk1217@inha.ac.kr (W.-K.R.); khj690114@hanmail.net (H.-J.K.); 2Department of Radiology, Inha University Hospital, Inha University College of Medicine, Incheon 22332, Republic of Korea; pengoon@gmail.com

**Keywords:** brain metastasis, bone metastasis, spine metastasis, cerebrospinal fluid space, non-small-cell lung cancer

## Abstract

*Background and Objectives*: The mechanisms involved in the development of brain metastasis (BM) remain elusive. Here, we investigated whether BM is associated with spine involvement in patients with non-small-cell lung cancer (NSCLC). *Materials and Methods*: A consecutive 902 patients with metastatic NSCLC were included from the Inha Lung Cancer Cohort. Patients with BM at diagnosis or subsequent BM development were evaluated for both spine involvement in NSCLC and anatomic proximity of BM to the cerebrospinal fluid (CSF) space. *Results:* At diagnosis, BM was found in 238 patients (26.4%) and bone metastasis was found in 393 patients (43.6%). In patients with bone metastasis, spine involvement was present in 280 patients. BM subsequently developed in 82 (28.9%) of 284 patients without BM at diagnosis. The presence of spine metastasis was associated with BM at diagnosis and subsequent BM development (adjusted odd ratios and 95% confidence intervals = 2.42 and 1.74–3.37, *p* < 0.001; 1.94 and 1.19–3.18, *p* = 0.008, respectively). Most patients with spine metastasis, either with BM at diagnosis or subsequent BM, showed BM lesions located adjacent (within 5mm) to the CSF space (93.8% of BM at the diagnosis, 100% of subsequent BM). *Conclusions:* These findings suggest that the presence of spine involvement is a risk factor for BM development in NSCLC patients with bone metastasis.

## 1. Introduction

Brain metastasis (BM) occurs in 20% to 40% of patients with advanced non-small-cell lung cancer (NSCLC) [1,2]. The early detection of BM is of clinical interest as its presence often confers a dismal prognosis with a median survival of 6 months and a deterioration of quality of life in NSCLC patients [3]. Imaging to detect BM is routinely recommended for newly diagnosed NSCLC patients, regardless of symptoms. Yet, there is no clear guidance on BM surveillance in patients without BM at diagnosis. The identification of high-risk groups for BM is an unmet clinical need.

The most common organs to which NSCLC metastasizes are bone and brain [4]. In particular, the spine is commonly involved in NSCLC patients with bone metastasis. Several mechanisms for the development of BM have been suggested. Commonly, BM develops within the brain parenchyma, especially in watershed areas through hematogenous spread and disruption of the blood–brain barrier [5]. In other cases, BM can arise from the infiltration of cancer cells into the cerebrospinal fluid (CSF) through direct invasion from adjacent tumors or metastatic lesions in the spine [6]. Furthermore, cancer cells within the CSF could then invade the brain parenchyma through the disruption of the blood–CSF barrier in patients with leptomeningeal metastasis (LM) [7].

In a previous study on patients with breast cancer, patients with bone metastases showed a high frequency of subsequent brain metastases [8]. In addition, a recent study showed that there were groups of patients diagnosed with incidental LM by CSF tapping among patients with solid tumors receiving spinal stereotactic radiotherapy for spine metastases [9]. However, there is little clinical evidence to support BM through the CSF pathway from the spine. There has been speculation that metastases in the spine could potentially enter the CSF through the retrograde spread of cancer cells along the valveless venous plexus encircling the vertebral column, but this potential metastatic route remains hypothetical [10,11].

Therefore, we hypothesized that patients with metastatic involvement of the spine could have an increased risk for BM. We investigated whether the presence of spine involvement was associated with BM in NSCLC patients.

## 2. Materials and Methods

### 2.1. Study Population

A total of 902 consecutive patients diagnosed with stage IV NSCLC between January 2005 and December 2018 at Inha University Hospital (Incheon, Republic of Korea) were initially considered for this study. All patients underwent computed tomography (CT) of the chest and upper abdomen, ^18^F-fluorodeoxyglucose positron emission tomography/computed tomography (FDG-PET/CT) scan, and brain magnetic resonance imaging (MRI) at diagnosis and follow-up. Information such as gender, smoking history, Eastern Cooperative Oncology Group (ECOG) performance status, histology, mutational status of the epidermal growth factor receptor (EGFR) gene, T category, N category, and organs of metastasis were analyzed. The stage of all patients was estimated according to the eighth edition of the TNM classification system [12]. All information was collected prospectively from the Inha Lung Cancer Cohort (ILCC) [13]. This study was approved by the Institutional Review Board of Inha University Hospital (2020-03-018) and informed consent was obtained from patients.

### 2.2. Identification of Brain or Bone Metastasis and Spine Involvement

BM was identified based on brain MRI. In patients without BM at diagnosis, subsequent BM development was evaluated. In patients with BM at diagnosis, the presence of metastatic lesions within 5 mm of the CSF space was investigated [14]. The presence of intracranial LM was also evaluated. All imaging was reviewed by a radiologist. Bone metastasis was identified based on FDG-PET/CT scan results. All PET images were corrected for attenuation using the acquired CT data. The presence of abnormal FDG uptake was indicated when the accumulation of the radiotracer moderately-to-markedly increased relative to the expected uptake in normal structures or surrounding tissue, with the exclusion of physiologic bowel and urinary activity. Bone metastases were classified into spine and non-spine involvement by their location. Spinal canal invasion on spine MRIs was evaluated.

### 2.3. Statistical Analysis

The association between BM and clinical variables was assessed using chi-square tests. Univariate and multivariate binary logistic regression analyses were performed to identify the association of spine metastasis with BM at diagnosis along with odds ratios (ORs) and 95% confidence intervals (CIs). To assess the effect of spine metastasis on subsequent BM, we performed univariate and multivariate analyses using the Cox proportional hazards model. Variables that were found to have a value of *p* ≤ 0.1 in univariate analysis were included in a multivariate Cox proportional hazards model. Statistical significance was considered as two-sided *p* values of ≤0.05. All statistical analyses were performed using a statistical software package (SPSS, version 19.0, Chicago, IL, USA).

## 3. Results

### 3.1. Patient Characteristics by BM at Diagnosis or Subsequent BM

The median age of the 902 patients was 69 years (range, 34–96) (Table 1). At diagnosis, 238 patients (26.4%) had BM and 393 patients (43.6%) had bone metastasis. The distributions of age, ECOG performance status, T category, and N category were not different between patients with and without BM. However, BM was common in female patients (*p* = 0.015), those who had never smoked (*p* < 0.001), and patients with adenocarcinoma histology (*p* = 0.003), EGFR activating mutations (*p* = 0.013), and bone metastasis (*p* < 0.001). Among the patients with bone metastasis, metastatic involvement of the spine was present in 280 patients (71.2%) and common in those with BM (*p* < 0.001).

Two hundred and eighty-four patients without BM at diagnosis were followed up for subsequent BM development with serial brain MRIs (interval of follow-up, median and 95% CI = 6.6 months and 4.1–8.3 months) (Table 2). Of these, subsequent BM was observed in 82 patients (28.9%). Subsequent BM was more common in female patients (*p* = 0.003), patients with an age ≤ 69 (*p* = 0.034), and those who had never smoked (*p* = 0.008). Furthermore, subsequent BM was more common in patients with adenocarcinoma histology (*p* = 0.001), EGFR-activating mutations (*p* < 0.001), and bone metastases at diagnosis, especially with spine involvement (*p* < 0.001).

### 3.2. Association of Spine Metastasis with BM at Diagnosis or Subsequent BM

Female gender (OR and 95% CI = 1.46 and 1.08–1.98, *p* = 0.015), never having smoked (1.77 and 1.30–2.40, *p* < 0.001), adenocarcinoma histology (1.91 and 1.27–2.87, *p* = 0.002), higher N category (1.55 and 1.06–2.29, *p* = 0.026), and bone metastasis (2.31 and 1.71–3.12, *p* < 0.001) were significantly associated with an increased risk of BM at diagnosis (Table 3 and Appendix A). Bone metastasis showed a significant association with BM after adjustment for potential confounding by other clinical variables (2.07 and 1.50–2.84, *p* < 0.001). Spine involvement was significantly associated with the risk of BM in multivariate logistic regression analysis (2.42 and 1.74–3.37, *p* < 0.001), but non-spine involvement was not (0.89 and 0.55– 1.42, *p* = 0.615) (Appendix A).

Higher N category (2.47 and 1.45–4.21, *p* = 0.001) and spine involvement (2.46 and 1.53–3.94, *p* < 0.001) were significantly associated with an increased risk of subsequent BM development (Table 4). Spine involvement showed a significant association with subsequent BM development after adjustment for potential confounding by other clinical variables (1.94 and 1.19–3.18, *p* = 0.008) (Figure 1).

### 3.3. Anatomic Proximity of BM Lesions to CSF Space in Patients with Spine Metastasis

BM lesions adjacent (within 5 mm) to the CSF space were observed in 105 (93.8%) of 112 patients with both spine metastasis and BM at diagnosis and in 26 (100%) of 26 patients with spine metastasis at diagnosis and subsequent BM development. In addition, intracranial LM on brain MRI was observed in 35.7% of patients with spine metastasis and BM at diagnosis and in 61.5% of patients with spine metastasis at diagnosis and subsequent BM development. Of the 33 patients who underwent spine MRI, spinal canal invasion was noted in 17 (65.4%) of 26 patients with BM at diagnosis, and 5 (71.4%) of 7 patients with subsequent BM development.

## 4. Discussion

This study, for the first time, demonstrated an association between spine involvement in bone metastasis with BM, either present at diagnosis or developed subsequently. Interestingly, the results suggest a sequential causal relationship through the anatomic proximities of the brain, spine, and CSF. In this study, the presence of spine involvement in bone metastasis was significantly associated with BM at diagnosis or subsequent BM development in the univariate analysis. These effects on BM were maintained after adjustments for potential confounders.

BM lesions adjacent to the CSF space were observed in most patients with spine metastasis and BM at diagnosis or subsequent BM [15]. In addition, spinal canal invasion on spine MRI was observed in a significant proportion of patients with spine metastasis and BM at the time of diagnosis. Furthermore, intracranial LM on brain MRI was commonly observed in these patients. These findings support the fact that spine metastasis is present with BM at diagnosis or with subsequent BM development. Taken together, the data suggest that cancer cells in the spine could metastasize to the brain via the CSF [6]. Our study suggests the need for prophylactic spine radiotherapy before spinal canal involvement and subsequent LM and BM occur due to spinal metastatic lesions. A study with this design has never been conducted before, and clinical trials regarding this topic are needed in the future.

In a previous study with 592 NSCLC patients, 59 of 102 patients (57.8%) with LM had concurrent BM at diagnosis. In addition, patients with LM had a significantly high rate of bone metastases compared to patients with only BM or no CNS metastases, which is in line with our study. In another study of 125 non-small-cell lung cancer patients with LM, 102 (82%) patients had brain metastases [16]. Consistent with prior studies, a significant proportion of patients in our study had coexisting BM and LM. In addition, spinal canal invasion was observed by spinal MRI in a significant proportion of patients with small-cell lung cancer in a study with 163 solid tumor patients with LM [17]. This supports the results of our study, which emphasized the mechanism of LM development through tumor invasion of the spinal canal.

Circulating tumor cells (CTCs) present in the CSF and leptomeninges pose significant challenges to therapeutic interventions and have the potential to initiate metastasis in the brain and spine [18]. A considerable proportion of CNS metastases affecting the leptomeninges originate from breast and lung tumors. Exploiting the nutrient-poor microenvironment of the CSF, tumor cells adapt to enhance their survival. A prior investigation revealed that invasive tumor cells release complement C3 into the CSF, which then interacts with the C3aR receptor on choroid plexus cells. This interaction disrupts the blood–CSF barrier (BCSFB), enabling the unimpeded entry of nutrients and growth factors into the CSF. Further exploration is warranted to determine whether BCSFB disruption facilitates the invasion of tumors into the brain parenchyma.

It is important to measure the impact that brain metastases have on quality of life (QoL) for a complete picture of the disease burden [19]. Symptoms of BM include headaches, cognitive deficits, ataxia, seizures, and visual and speech problems, which can impact patient’s QoL in addition to the symptoms from their primary tumor. Furthermore, the side effects associated with the treatment for BM can seriously impact a patient’s QoL by limiting their ability to perform everyday activities and by altering neurocognitive processes, especially if the treatment involves surgery or radiation to the brain. Therefore, it is important to identify high-risk groups for BM in advance, and this study provides useful information in this regard.

Patients that were EGFR-positive or ALK-positive had higher rates of BM than those with wild-type tumors, which is supported by previous studies [20]. The blood–brain barrier shields the CNS from harmful substances, yet simultaneously hinders the majority of therapeutic agents from reaching the brain parenchyma and leptomeningeal space [21]. The influx and efflux mechanisms collectively determine the entry of drugs into the CSF. Active efflux transport systems act as barriers, impeding the delivery of drugs to the CNS. Export proteins, like P-glycoprotein and the breast cancer resistance protein, situated in the luminal membrane of the brain capillary endothelium, are the primary obstacles to efficient drug transport into the brain and leptomeningeal space. While various anticancer treatments, including tyrosine kinase inhibitors and chemotherapeutic agents, serve as substrates for these efflux transporters, their penetration into the CSF varies due to the opposing interplay of influx and efflux mechanisms. For instance, despite being a substrate for efflux transporters, osimertinib, a third-generation EGFR tyrosine kinase inhibitor, exhibits adequate permeability to counteract efflux [22]. Alectinib, a second-generation inhibitor targeting ALK and known for its remarkable ability to penetrate the CNS, demonstrates notable effectiveness in both systemic and CNS outcomes for individuals with ALK-rearranged NSCLC [23,24]. The landscape of NSCLC treatment has been significantly altered by the advent of immunotherapy. Despite their substantial impact, programmed death-1/PD-ligand 1 pathway inhibitors face challenges in penetrating the blood–brain barrier due to their high molecular weight [25]. Instead, they exert their effects by systemically activating immune cells. Potential avenues for access to the CSF compartment and brain tissues include the choroid plexus and CSF, enabling peripheral immune cells and large molecules to enter. AZD3759, a recently developed EGFR tyrosine kinase inhibitor of the new generation, exhibits encouraging efficacy in an EGFR-mutant mouse model with LM originating from NSCLC [22]. Lorlatinib, a next-generation ALK inhibitor, was specifically designed to reduce drug efflux facilitated by P-glycoprotein [26].

A recent study demonstrated that vertebral skeletal stem cells (vSSC) are distinct from other skeletal stem cells and mediate the unique physiology and pathology of vertebrae, including contributing to the high rate of vertebral metastasis [27]. In particular, human vSSCs secreting MFGE8 were more likely to interact with cancer cells than were those vSSCs that were not secreting the protein. Considering our results, we suggest that treatment targeting vSSC and MFGE8 has the potential to prevent the development of not only spinal metastases but also brain metastases.

There are several limitations in this study. First, the recruitment of patients from only a single center challenges the generalizability of these results and necessitates external confirmatory studies. However, clinical information was prospectively obtained from the ILCC and imaging studies were extensively evaluated by an experienced neuro-radiologist. Second, treatments given to NSCLC patients were not considered in this study and their effects on the development of BM remain elusive. Finally, the presence of LM was not confirmed with cytological examination of CSF in most patients due to the difficulty of CSF tapping. Alternatively, the proportion of patients with LM on brain MRI was analyzed. Within the limits of the study, the results suggest CSF is a potential pathway of spine involvement in bone metastasis to BM.

## 5. Conclusions

In conclusion, this study suggests that spine metastases at diagnosis is a risk factor for baseline and subsequent BM development in patients with NSCLC. As such, clinicians should carefully monitor subsequent BM with brain MRI in NSCLC patients with spine involvement.

## Figures and Tables

**Figure 1 medicina-60-00152-f001:**
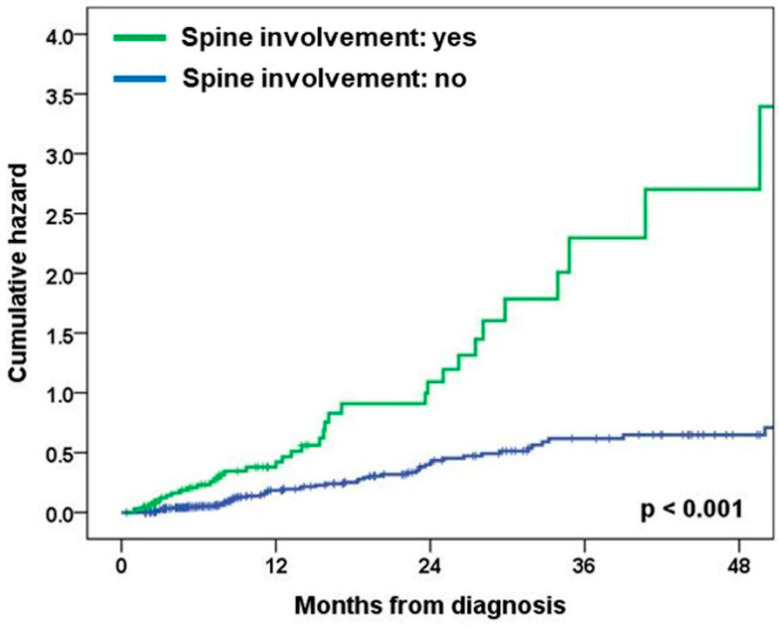
Effect of spine involvement in bone metastasis on subsequent BM development.

**Table 1 medicina-60-00152-t001:** Patient characteristics for BM.

Variables	Brain Metastasis		
	Yes (*n* = 238)	No (*n* = 664)	*p* Value
Age, median			0.108
>69	106 (44.5)	336 (50.6)	
≤69	132 (55.5)	328 (49.4)	
Gender			0.015
Male	137 (57.6)	441 (66.4)	
Female	101 (42.4)	223 (33.6)	
Smoking history			<0.001
Ever	138 (58.2)	465 (71.1)	
Never	99 (41.8)	189 (28.9)	
ECOG performance status			0.057
0–1	132 (56.2)	416 (63.2)	
≥2	103 (43.8)	242 (36.8)	
Histology			0.003
SQC	34 (14.3)	154 (23.2)	
ADC	188 (79.0)	447 (67.3)	
Others	16 (6.7)	63 (9.5)	
EGFR mutation			0.013
Negative	147 (61.8)	468 (70.5)	
Positive	91 (38.2)	196 (29.5)	
T category			0.403
Tx	2 (0.8)	16 (2.4)	
T1	9 (3.8)	38 (5.7)	
T2	38 (16.0)	113 (17.0)	
T3	53 (22.3)	139 (20.9)	
T4	136 (57.1)	358 (53.9)	
N category			0.145
N0	47 (19.7)	180 (27.1)	
N1	21 (8.8)	60 (9.0)	
N2	54 (22.7)	137 (20.7)	
N3	116 (48.7)	286 (43.1)	
M category			<0.001
Others	98 (41.2)	411 (61.9)	
Bone	140 (58.8)	253 (38.1)	
Non-spine	28 (20.0)	85 (33.6)	
Spine	112 (80.0)	168 (66.4)	

ECOG, Eastern Cooperative Oncology Group; SQC, squamous cell carcinoma, ADC, adenocarcinoma; EGFR, epidermal growth factor receptor.

**Table 2 medicina-60-00152-t002:** Patient characteristics for subsequent BM.

Variables	Subsequent Brain Metastasis		
	Yes (*n* = 82)	No (*n* = 202)	*p* Value
Age, median			0.034
>69	25 (30.5)	89 (44.1)	
≤69	57 (69.5)	113 (55.9)	
Gender			0.003
Male	41 (50.0)	139 (68.8)	
Female	41 (50.0)	63 (31.2)	
Smoking history			0.008
Ever	46 (56.1)	146 (72.3)	
Never	36 (43.9)	56 (27.7)	
ECOG performance status			0.863
0–1	57 (71.2)	146 (72.3)	
≥2	23 (28.8)	56 (27.7)	
Histology			0.001
SQC	6 (7.3)	49 (24.3)	
ADC	72 (87.8)	134 (66.3)	
Others	4 (4.9)	19 (9.4)	
EGFR mutation			<0.001
Negative	38 (46.3)	147 (72.8)	
Positive	44 (53.7)	55 (27.2)	
T category			0.879
Tx	3 (3.7)	4 (2.0)	
T1	6 (7.3)	14 (6.9)	
T2	13 (15.9)	40 (19.8)	
T3	17 (20.7)	41 (20.3)	
T4	43 (52.4)	103 (51.0)	
N category			0.109
N0	21 (25.6)	72 (35.6)	
N1	5 (6.1)	23 (11.4)	
N2	17 (20.7)	29 (14.4)	
N3	39 (47.6)	78 (38.6)	
Bone metastasis			0.001
No	42 (51.2)	148 (73.3)	
Yes	40 (48.8)	54 (26.7)	
Non-spine	14 (35.0)	22 (40.7)	
Spine	26 (65.0)	32 (59.3)	

ECOG, Eastern Cooperative Oncology Group; SQC, squamous cell carcinoma, ADC, adenocarcinoma; EGFR, epidermal growth factor receptor.

**Table 3 medicina-60-00152-t003:** Association of spine involvement of bone metastasis with BM at diagnosis.

Variables	Univariate Analysis		Multivariate Analysis	
	OR (95% CI)	*p* Value	OR (95% CI)	*p* Value
Age, median				
>69	reference	-		
≤69	1.28 (0.95–1.72)	0.109		
Gender				
Male	reference	-	reference	-
Female	1.46 (1.08–1.98)	0.015	0.77 (0.44–1.36)	0.371
Smoking history		-		
Ever	reference		reference	-
Never	1.77 (1.30–2.40)	<0.001	2.01 (1.12–3.60)	0.020
ECOG performance status				
0–1	reference	-	reference	-
≥2	1.34 (0.99–1.82)	0.057	1.26 (0.91–1.73)	0.160
Histology				
SQC	reference	-	reference	-
ADC	1.91 (1.27–2.87)	0.002	1.53 (0.97–2.42)	0.067
Others	1.15 (0.59–2.23)	0.679	1.11 (0.56–2.19)	0.763
EGFR mutation				
Negative	reference	-	reference	-
Positive	1.48 (1.08–2.02)	0.014	0.92 (0.63–1.33)	0.643
T category				
Tx–T1	reference	-	reference	-
T2	1.65 (0.78–3.48)	0.187	1.49 (0.69–3.24)	0.310
T3	1.87 (0.91–3.85)	0.089	1.61 (0.75–3.42)	0.219
T4	1.87 (0.95–3.67)	0.072	1.49 (0.74–3.04)	0.268
N category				
N0	reference	-	reference	-
N1	1.34 (0.74–2.42)	0.332	1.58 (0.85–2.94)	0.150
N2	1.51 (0.96–2.37)	0.073	1.78 (1.10–2.86)	0.019
N3	1.55 (1.06–2.29)	0.026	1.42 (0.93–2.15)	0.105
Bone metastasis				
Non-spine	reference	-	reference	-
Spine	2.62 (1.93–3.57)	<0.001	2.42 (1.74–3.37)	<0.001

ECOG, Eastern Cooperative Oncology Group; SQC, squamous cell carcinoma, ADC, adenocarcinoma; EGFR, epidermal growth factor receptor.

**Table 4 medicina-60-00152-t004:** Effect of spine involvement of bone metastasis on subsequent BM.

Variables	Univariate Analysis		Multivariate Analysis	
	HR (95% CI)	*p* Value	HR (95% CI)	*p* Value
Age, median				
>69	reference	-		
≤69	1.15 (0.72–1.84)	0.566		
Gender				
Male	reference	-		
Female	1.30 (0.84–2.01)	0.238		
Smoking history				
Ever	reference	-		
Never	1.19 (0.77–1.84)	0.440		
ECOG performance status				
0–1	reference	-		
≥2	1.32 (0.81–2.16)	0.260		
Histology				
SQC	reference	-	reference	-
ADC	2.02 (0.88–4.67)	0.099	2.07 (0.86–5.01)	0.106
Others	2.29 (0.65–8.14)	0.200	2.36 (0.64–8.72)	0.197
EGFR mutation				
Negative	reference	-		
Positive	1.34 (0.87–2.07)	0.188		
T category				
Tx–T1	reference	-		
T2	0.67 (0.28–1.57)	0.352		
T3	1.02 (0.45–2.29)	0.966		
T4	0.92 (0.45–1.90)	0.830		
N category				
N0	reference	-	reference	-
N1	1.46 (0.55–3.89)	0.451	1.83 (0.66–5.07)	0.243
N2	3.03 (1.58–5.78)	0.001	2.99 (1.54–5.82)	0.001
N3	2.47 (1.45–4.21)	0.001	2.32 (1.35–3.98)	0.002
Bone metastasis				
Non-spine	reference	-	reference	-
Spine	2.46 (1.53–3.94)	<0.001	1.94 (1.19–3.18)	0.008

ECOG, Eastern Cooperative Oncology Group; SQC, squamous cell carcinoma, ADC, adenocarcinoma; EGFR, epidermal growth factor receptor.

## Data Availability

The data presented in this study are available in the manuscript. Additional raw data are available on request from the corresponding author.

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
