# Peer review of "Spine Metastasis Is Associated with the Development of Brain Metastasis in Non-Small-Cell Lung Cancer Patients"

_medicina, 2024, doi:10.3390/medicina60010152_

Round 1
Reviewer 1 Report
Comments and Suggestions for Authors
In the manuscript entitled “Spine metastasis is associated with the development of brain metastasis in non-small-cell lung cancer patients” the authors try to prove that brain metastases may be associated with spine metastases in NSCLC patients.
Introduction
The introduction section is too short, therefore it is impossible for a reader to immerse in the topic. Moreover, the section lacks a clear depiction of the aim of the study and its justification.
Materials and Methods
In section 2.2 authors mention the name of a researcher who reviewed the data. Co-authors should not be mentioned in the text by name
Results
Results are presented concisely and clearly.
Figure 1. The title should be located below the figure.
Discussion
The section is too short and the obtained results are not discussed with similar studies. The section should be rewritten, as in this state it is unacceptable.
Author Response
Reviewer 1
In the manuscript entitled “Spine metastasis is associated with the development of brain metastasis in non-small-cell lung cancer patients” the authors try to prove that brain metastases may be associated with spine metastases in NSCLC patients.
- Introduction
The introduction section is too short, therefore it is impossible for a reader to immerse in the topic. Moreover, the section lacks a clear depiction of the aim of the study and its justification.
→ Thank you for the kind suggestion. With respect to the reviewer’s comment, we added sentences into Introduction as following.
In INTRODUCTION:
“In other cases, BM can arise from the infiltration of cancer cells into the cerebrospinal fluid (CSF) through direct invasion from adjacent tumors or metastatic lesions in the spine [6]. Furthermore, cancer cells within the CSF could then invade the brain parenchyma through disruption of the blood-CSF barrier in patients with leptomeningeal metastasis (LM) [7].
In a previous study on patients with breast cancer, patients with bone metastases showed a high frequency of subsequent brain metastases [8]. In addition, a recent study showed that there were group of patients diagnosed with incidental LM by CSF tapping among patients with solid tumor receiving spinal stereotactic radiotherapy for spine metastases [9]. However, there is little clinical evidence to support BM through the CSF pathway from the spine. There had been a speculation that metastases in the spine could potentially enter the CSF through the retrograde spread of cancer cells along the valveless venous plexus encircling the vertebral column, but this potential metastatic route remains hypothetical [10,11].”
- Chowdhary, S.; Chamberlain, M.J.J.o.t.N.C.C.N. Leptomeningeal metastases: current concepts and management guidelines. 2005, 3, 693-703.
- Boire, A.; Zou, Y.; Shieh, J.; Macalinao, D.G.; Pentsova, E.; Massagué, J.J.C. Complement component 3 adapts the cerebrospinal fluid for leptomeningeal metastasis. 2017, 168, 1101-1113. e1113.
- Fujii, T.; Mason, J.; Chen, A.; Kuhn, P.; Woodward, W.A.; Tripathy, D.; Newton, P.K.; Ueno, N.T. Prediction of bone metastasis in inflammatory breast cancer using a markov chain model. The Oncologist 2019, 24, 1322-1330.
- Freret, M.E.; Wijetunga, N.A.; Shamseddine, A.A.; Higginson, D.S.; Schmitt, A.M.; Yamada, Y.; Lis, E.; Boire, A.; Yang, J.T.; Xu, A.J. Early Detection of Leptomeningeal Metastases Among Patients Undergoing Spinal Stereotactic Radiosurgery. Advances in Radiation Oncology 2023, 8, 101154.
- Bubendorf, L.; Schöpfer, A.; Wagner, U.; Sauter, G.; Moch, H.; Willi, N.; Gasser, T.C.; Mihatsch, M.J. Metastatic patterns of prostate cancer: an autopsy study of 1,589 patients. Human pathology 2000, 31, 578-583.
- Geldof, A.A. Models for cancer skeletal metastasis: a reappraisal of Batson's plexus. Anticancer research 1997, 17, 1535-1539.
- Materials and Methods
In section 2.2 authors mention the name of a researcher who reviewed the data. Co-authors should not be mentioned in the text by name
→ Thanks for your comment. With respect to the reviewer’s comment, the name of the co-author mentioned in the manuscript has been removed as following.
In Materials and Methods:
“The presence of intracranial LM was also evaluated. All imaging was reviewed by a radiologist (Lee HY).”
- Results
Results are presented concisely and clearly.
→ Thanks for your comment.
- Figure 1. The title should be located below the figure.
→ Thanks for your comment. With respect to the reviewer’s comment, the location of the title of Figure 1 was relocated below the figure.
- Discussion
The section is too short and the obtained results are not discussed with similar studies. The section should be rewritten, as in this state it is unacceptable.
→ Thank you for the kind suggestion. With respect to the reviewer’s comment, we added sentences into Discussion as following.
In DISCUSSION:
“In a previous study with 592 NSCLC patients, 59 of 102 patients (57.8%) with LM had concurrently BM at diagnosis [15]. In addition, patients with LM had a significantly high rate of bone metastases compared to patients with only BM or no CNS metastases, which is in line with our study. In another study of 125 non-small cell lung cancer patients with LM, 102 (82%) patients had brain metastases [16]. Consistent with prior studies, a significant proportion of patients in our study had coexisting BM and LM. In addition, spinal canal invasion was observed by spinal MRI in significant proportion of patients with small cell lung cancer in a study with 163 solid tumor patients with LM [17]. This supports the results of our study which emphasized the mechanism of LM development through tumor invasion of the spinal canal.”
- Li, Q.; Lin, Z.; Hong, Y.; Fu, Y.; Chen, Y.; Liu, T.; Zheng, Y.; Tian, J.; Liu, C.; Pu, W. Brain parenchymal and leptomeningeal metastasis in non-small cell lung cancer. Scientific Reports 2022, 12, 22372.
- Morris, P.G.; Reiner, A.S.; Szenberg, O.R.; Clarke, J.L.; Panageas, K.S.; Perez, H.R.; Kris, M.G.; Chan, T.A.; DeAngelis, L.M.; Omuro, A.M. Leptomeningeal metastasis from non-small cell lung cancer: survival and the impact of whole brain radiotherapy. Journal of Thoracic Oncology 2012, 7, 382-385.
- Pan, Z.; Yang, G.; He, H.; Yuan, T.; Wang, Y.; Li, Y.; Shi, W.; Gao, P.; Dong, L.; Zhao, G. Leptomeningeal metastasis from solid tumors: clinical features and its diagnostic implication. Scientific reports 2018, 8, 10445.

Reviewer 2 Report
Comments and Suggestions for Authors
In the article "Spine metastasis is associated with the development of brain metastasis in non-small-cell lung cancer patients" he analyzes in detail the relationship between spine metastatic involvement having a starting point in lung cancers and the risk of developing brain metastases. Also, the development of metastases adjacent to the cerebrospinal fluid is identified. Including tables and graphical representations, the article is innovative and very well written. I would propose to the authors a supplement with future perspectives in the context of therapies that cross the blood brain barrier (BBB) - for example Erlotinb, Crizotinib, Osimertinb, but also in the context of long survivals associated with immunotherapy. Are there prospects for prophylactic radiotherapy in the vicinity of CSF for these cases? it would be worth a discussion with arguments for and against. I would also suggest including more references.
Author Response
Reviewer 2
In the article "Spine metastasis is associated with the development of brain metastasis in non-small-cell lung cancer patients" he analyzes in detail the relationship between spine metastatic involvement having a starting point in lung cancers and the risk of developing brain metastases. Also, the development of metastases adjacent to the cerebrospinal fluid is identified. Including tables and graphical representations, the article is innovative and very well written.
- I would propose to the authors a supplement with future perspectives in the context of therapies that cross the blood brain barrier (BBB) - for example Erlotinb, Crizotinib, Osimertinb, but also in the context of long survivals associated with immunotherapy.
→ Thank you for the kind suggestion. With respect to the reviewer’s comment, we added sentences into Discussion as following.
In DISCUSSION:
“Patients with EGFR-positive or ALK-positive had higher rates of BM than wild-type tumor, which is supported by the previous studies [20]. The blood-brain barrier shields the CNS from harmful substances, yet simultaneously hinders the majority of therapeutic agents from reaching the brain parenchyma and leptomeningeal space [21]. The influx and efflux mechanisms collectively determine the entry of drugs into the CSF. Active efflux transport systems act as barriers, impeding the delivery of drugs to the CNS. Export proteins like P-glycoprotein and breast cancer resistance protein, situated in the luminal membrane of brain capillary endothelium, pose the primary obstacles to efficient drug transport into the brain and leptomeningeal space. While various anticancer treatments, including tyrosine kinase inhibitors and chemotherapeutic agents, serve as substrates for these efflux transporters, their penetration into the CSF varies due to the opposing interplay of influx and efflux mechanisms. For instance, despite being a substrate for efflux transporters, osimertinib, a third-generation EGFR tyrosine kinase inhibitor, exhibits ad-equate permeability to counteract efflux [22]. Alectinib, a second-generation inhibitor tar-geting ALK and known for its remarkable ability to penetrate the CNS, demonstrates no-table effectiveness in both systemic and CNS outcomes for individuals with ALK-rearranged NSCLC [23,24]. The landscape of NSCLC treatment has been significantly altered by the advent of immunotherapy. Despite their substantial impact, programmed death-1/PD-ligand 1 pathway inhibitors face challenges in penetrating the blood–brain barrier due to their high molecular weight [25]. Instead, they exert their effects by systemically activating immune cells. Potential avenues for access to the CSF compartment and brain tissues include the choroid plexus and CSF, enabling peripheral immune cells and large molecules to enter. AZD3759, a recently developed EGFR tyrosine kinase inhibitor of the new generation, exhibits encouraging efficacy in an EGFR-mutant mouse model with LM originating from NSCLC [22]. Lorlatinib, a next-generation ALK inhibitor, was specifically designed to reduce drug efflux facilitated by P-glycoprotein [26].”
- Gillespie, C.S.; Mustafa, M.A.; Richardson, G.E.; Alam, A.M.; Lee, K.S.; Hughes, D.M.; Escriu, C.; Zakaria, R. Genomic alterations and the incidence of brain metastases in advanced and metastatic non-small cell lung cancer: a systematic review and meta-analysis. Journal of Thoracic Oncology 2023.
- Cheng, H.; Perez-Soler, R. Leptomeningeal metastases in non-small-cell lung cancer. The Lancet Oncology 2018, 19, e43-e55.
- Ballard, P.; Yates, J.W.; Yang, Z.; Kim, D.-W.; Yang, J.C.-H.; Cantarini, M.; Pickup, K.; Jordan, A.; Hickey, M.; Grist, M. Preclinical comparison of osimertinib with other EGFR-TKIs in EGFR-mutant NSCLC brain metastases models, and early evidence of clinical brain metastases activity. Clinical Cancer Research 2016, 22, 5130-5140.
- Gainor, J.F.; Sherman, C.A.; Willoughby, K.; Logan, J.; Kennedy, E.; Brastianos, P.K.; Chi, A.S.; Shaw, A.T. Alectinib salvages CNS relapses in ALK-positive lung cancer patients previously treated with crizotinib and ceritinib. Journal of Thoracic Oncology 2015, 10, 232-236.
- Gainor, J.F.; Chi, A.S.; Logan, J.; Hu, R.; Oh, K.S.; Brastianos, P.K.; Shih, H.A.; Shaw, A.T. Alectinib dose escalation reinduces central nervous system responses in patients with anaplastic lymphoma kinase–positive non–small cell lung cancer relapsing on standard dose alectinib. Journal of Thoracic Oncology 2016, 11, 256-260.
- O’Kane, G.M.; Leighl, N.B. Are immune checkpoint blockade monoclonal antibodies active against CNS metastases from NSCLC?—current evidence and future perspectives. Translational Lung Cancer Research 2016, 5, 628.
- Johnson, T.W.; Richardson, P.F.; Bailey, S.; Brooun, A.; Burke, B.J.; Collins, M.R.; Cui, J.J.; Deal, J.G.; Deng, Y.-L.; Dinh, D. Discovery of (10 R)-7-Amino-12-fluoro-2, 10, 16-trimethyl-15-oxo-10, 15, 16, 17-tetrahydro-2H-8, 4-(metheno) pyrazolo [4, 3-h][2, 5, 11]-benzoxadiazacyclotetradecine-3-carbonitrile (PF-06463922), a macrocyclic inhibitor of anaplastic lymphoma kinase (ALK) and c-ros oncogene 1 (ROS1) with preclinical brain exposure and broad-spectrum potency against ALK-resistant mutations. Journal of medicinal chemistry 2014, 57, 4720-4744.
- Are there prospects for prophylactic radiotherapy in the vicinity of CSF for these cases? It would be worth a discussion with arguments for and against.
→ Thank you for the kind suggestion. With respect to the reviewer’s comment, we added sentences into Discussion as following.
In DISCUSSION:
“Our study suggests the need for prophylactic spine radiotherapy before spinal canal involvement and subsequent LM and BM occur due to spinal metastatic lesions. A study with this design has never been conducted before, and clinical trials about this topic is needed in the future.”
- I would also suggest including more references.
→ Thank you for the kind suggestion. With respect to the reviewer’s comment, we have added several references in the revised manuscript.

Reviewer 3 Report
Comments and Suggestions for Authors
1. Metastatic involvement of the spine and the risk for brain metastasis
2. The topic is original, and address a gap in the literature regarding.
3. Metastasis in individuals already diagnosed with brain metastatis, possible mechanistic explanation.
4. No specific improvements should the authors consider regarding the methodology
5. The authors should improve the discussion as the comments provided below.
6. the references are appropriate.
7. No additional comments. on the tables and figures.
---------------------------------------------
There are two major concerns in the mansucript.
1) Discussion should be improved.
a) Discuss why did the authors hypothesized at the first place regarding CSF flow metastasis. Please, provide another hypothesis besides the CSF involvement.
b) Discuss the cumulative hazard assocaited with spine metastasis. What this reflects in terms of patient qualiy of life and why does this happens.
c) Talk more about the anatomic proximity between CSF and related metastasis, is there any possible receptor.
2) Conclusion. Please, be more descriptive in the conclusion, and how does this help.
Comments on the Quality of English Language
1. Metastatic involvement of the spine and the risk for brain metastasis
2. The topic is original, and address a gap in the literature regarding.
3. Metastasis in individuals already diagnosed with brain metastatis, possible mechanistic explanation.
4. No specific improvements should the authors consider regarding the methodology
5. The authors should improve the discussion as the comments provided below.
6. the references are appropriate.
7. No additional comments. on the tables and figures.
---------------------------------------------
There are two major concerns in the mansucript.
1) Discussion should be improved.
a) Discuss why did the authors hypothesized at the first place regarding CSF flow metastasis. Please, provide another hypothesis besides the CSF involvement.
b) Discuss the cumulative hazard assocaited with spine metastasis. What this reflects in terms of patient qualiy of life and why does this happens.
c) Talk more about the anatomic proximity between CSF and related metastasis, is there any possible receptor.
2) Conclusion. Please, be more descriptive in the conclusion, and how does this help.
Author Response
Reviewer 3
There are two major concerns in the mansucript.
1. Discussion should be improved.
a. Discuss why did the authors hypothesized at the first place regarding CSF flow metastasis. Please, provide another hypothesis besides the CSF involvement.
→ Thank you for the kind suggestion. With respect to the reviewer’s comment, we added sentences into Introduction as following.
In INTRODUCTION:
“In other cases, BM can arise from the infiltration of cancer cells into the cerebrospinal fluid (CSF) through direct invasion from adjacent tumors or metastatic lesions in the spine [6]. Furthermore, cancer cells within the CSF could then invade the brain parenchyma through disruption of the blood-CSF barrier in patients with leptomeningeal metastasis (LM) [7].
In a previous study on patients with breast cancer, patients with bone metastases showed a high frequency of subsequent brain metastases [8]. In addition, a recent study showed that there were group of patients diagnosed with incidental LM by CSF tapping among patients with solid tumor receiving spinal stereotactic radiotherapy for spine metastases [9]. However, there is little clinical evidence to support BM through the CSF pathway from the spine. There had been a speculation that metastases in the spine could potentially enter the CSF through the retrograde spread of cancer cells along the valveless venous plexus encircling the vertebral column, but this potential metastatic route remains hypothetical [10,11].”
b. Discuss the cumulative hazard associated with spine metastasis. What this reflects in terms of patient quality of life and why does this happens.
→ Thank you for the kind suggestion. With respect to the reviewer’s comment, we added sentences into Discussion as following.
In DISCUSSION:
“It is important to measure the impact that brain metastases have on quality of life (QoL) for a complete picture of the disease burden [19]. Symptoms of BM include headaches, cognitive deficits, ataxia, seizures and visual and speech problems, which can impact patient’s QoL in addition to the symptoms from their primary tumor. Furthermore, the side effects associated with treatment for BM can seriously impact a patient’s QoL by limiting their ability to perform everyday activities and by altering neurocognitive processes, especially if the treatment involves surgery or radiation to the brain. Therefore, it is important to identify high risk group for BM in advance, and this study provides useful information in this regard.”
- Peters, S.; Bexelius, C.; Munk, V.; Leighl, N. The impact of brain metastasis on quality of life, resource utilization and survival in patients with non-small-cell lung cancer. Cancer treatment reviews 2016, 45, 139-162.
c. Talk more about the anatomic proximity between CSF and related metastasis, is there any possible receptor.
→ Thank you for the kind suggestion. With respect to the reviewer’s comment, we added sentences into Discussion as following.
In DISCUSSION:
“Circulating tumor cells (CTCs) present in the CSF and leptomeninges pose significant challenges to therapeutic interventions and have the potential to initiate metastasis in the brain and spine [18]. A considerable proportion of CNS metastases affecting the leptomeninges originate from breast and lung tumors. Exploiting the nutrient-poor microenvironment of the CSF, tumor cells adapt to enhance their survival. A prior investigation revealed that invasive tumor cells release complement C3 into the CSF, which then interacts with the C3aR receptor on choroid plexus cells. This interaction disrupts the blood-CSF barrier (BCSFB), enabling the unimpeded entry of nutrients and growth factors into the CSF. Further exploration is warranted to determine whether BCSFB disruption facilitates the invasion of tumors into the brain parenchyma.”
- Deshpande, K.; Buchanan, I.; Martirosian, V.; Neman, J. Clinical perspectives in brain metastasis. Cold Spring Harbor Perspectives in Medicine 2019, a037051.
2. Conclusion. Please, be more descriptive in the conclusion, and how does this help.
→ Thank you for the kind suggestion. With respect to the reviewer’s comment, we revised sentences into Discussion as following.
In CONCLUSION:
“In conclusion, this study suggests that spine involvement metastases at diagnosis is a risk factor for baseline and subsequent BM development in patients with NSCLC. As such, clinicians should carefully monitor subsequent BM with brain MRI in NSCLC patients with spine involvement.”
